# Functional Role of Taurine in Aging and Cardiovascular Health: An Updated Overview

**DOI:** 10.3390/nu15194236

**Published:** 2023-09-30

**Authors:** Gaetano Santulli, Urna Kansakar, Fahimeh Varzideh, Pasquale Mone, Stanislovas S. Jankauskas, Angela Lombardi

**Affiliations:** 1Department of Medicine, Fleischer Institute for Diabetes and Metabolism (FIDAM), Einstein-Mount Sinai Diabetes Research Center (ES-DRC), Einstein Institute for Aging Research, Albert Einstein College of Medicine, New York, NY 10461, USA; urna.kansakar@einsteinmed.edu (U.K.); stanislovas.jankauskas@einsteinmed.edu (S.S.J.); angela.lombardi@einsteinmed.edu (A.L.); 2Department of Molecular Pharmacology, Division of Cardiology, Wilf Family Cardiovascular Research Institute, Albert Einstein College of Medicine, New York, NY 10461, USA; fahimeh.varzideh@einsteinmed.edu (F.V.); drpasquale.mone@gmail.com (P.M.)

**Keywords:** aging, 2-aminoethanesulfonic acid, cardiovascular risk, energy drinks, inflammation, metabolism, oxidative stress, supplements, tauric acid, taurine

## Abstract

Taurine, a naturally occurring sulfur-containing amino acid, has attracted significant attention in recent years due to its potential health benefits. Found in various foods and often used in energy drinks and supplements, taurine has been studied extensively to understand its impact on human physiology. Determining its exact functional roles represents a complex and multifaceted topic. We provide an overview of the scientific literature and present an analysis of the effects of taurine on various aspects of human health, focusing on aging and cardiovascular pathophysiology, but also including athletic performance, metabolic regulation, and neurological function. Additionally, our report summarizes the current recommendations for taurine intake and addresses potential safety concerns. Evidence from both human and animal studies indicates that taurine may have beneficial cardiovascular effects, including blood pressure regulation, improved cardiac fitness, and enhanced vascular health. Its mechanisms of action and antioxidant properties make it also an intriguing candidate for potential anti-aging strategies.

## 1. Introduction

Taurine (2-aminoethanesulfonic acid, also known as tauric acid) is a non-protein amino acid found in various animal tissues, especially in the brain, heart, and skeletal muscles. It is also present in several foods, such as meat, fish, dairy products, and energy drinks.

The main aim of this review is to summarize the key functional roles played by taurine in aging and in cardiovascular pathophysiology, especially based on the most recent findings in these fields. Specifically, taurine has been linked to, antioxidant activity, anti-inflammatory effects, and blood pressure regulation, with major implications for human health.

## 2. Nomenclature, Chemistry, and Biochemistry

The name taurine derives from the Latin taurus (cognate to Ancient Greek *ταῦρος*, “taûros”) meaning bull or ox: indeed, taurine was first isolated from the bile of the ox, *Bos taurus*, in 1827 by the German scientists Leopold Gmelin and Friedrich Tiedemann [1]. Early studies focused on its presence in animal tissues, where it was found in high concentrations in the brain, heart, and skeletal muscles. Later on, in 1846, the English chemist Edmund Ronalds confirmed the presence of taurine in human bile [2]. Taurine is detected in high concentrations in oxidative tissues, characterized by a high number of mitochondria, and in lower concentrations in glycolytic tissues [3,4,5,6]. The taurine content in various human tissues is reported in Table 1; over the years, researchers have explored its role in various physiological processes, leading to an increased understanding of its significance in human health.

Chemically, taurine is classified as a beta-amino acid, and its molecular formula is C_2_H_7_NO_3_S (Molecular Weight, MW: 125.15). Structurally, it is characterized by an amino group (NH_2_) and a sulfonic acid group (SO_3_H) attached to the beta carbon (Figure 1); unlike other amino acids, taurine lacks a chiral center, meaning it is optically inactive; its relatively simple structure allows it to perform diverse functions within the body.

While the human body can synthesize taurine to some extent, dietary intake is essential to maintain optimal levels. Foods rich in taurine include meat, fish, poultry, and dairy products. Vegetarians and vegans may have a lower taurine intake due to their dietary restrictions [12], but the significance of this in terms of deficiency remains unclear.

Taurine is synthesized in humans in the liver mainly via the “cysteine sulfinic pathway” (Figure 2). Cysteine dioxygenase oxidizes cysteine to form cysteine sulfinic acid, which is then decarboxylated by cysteine sulfinic acid decarboxylase to obtain hypotaurine, which is then oxidized by hypotaurine dioxygenase to form taurine [13,14,15,16,17,18]. An alternative pathway is trans-sulfuration, in which homocysteine is converted into cystathionine, which is then transformed into hypotaurine by cystathionine gamma-lyase, cysteine dioxygenase, and cysteine sulfinic acid decarboxylase, and finally oxidized to form taurine [19,20,21].

Taurine has been extensively studied to determine its effects on human health. In terms of cellular function, taurine is primarily found in the intracellular fluid of many tissues, where it plays a vital role in a number of physiological processes [22,23,24,25,26,27,28]. It acts as an osmolyte, regulating cell volume and maintaining cell integrity [29,30]. In the liver, taurine is conjugated with bile acids, forming bile salts that aid in fat digestion and absorption in the intestines [31,32,33,34]. These processes are crucial for lipid metabolism and absorption of fat-soluble vitamins [35].

Taurine has also been shown to be involved in calcium (Ca^2+^) signaling, modulation of ion channels, and neurotransmission, affecting neural excitability and synaptic transmission. Intriguingly, this amino acid exhibits important antioxidant properties, protecting cells from oxidative and nitrosative stress by scavenging free radicals and reactive oxygen species (ROS) [36,37,38,39,40,41,42,43,44]. These antioxidant actions certainly contribute to its potential benefits in terms of neuroprotection and cardiovascular health [45]. In fact, taurine is highly concentrated in the brain and several studies indicate that taurine might act as a neurotransmitter or neuromodulator, influencing neurotransmitter release and receptor function, affecting cognitive processes, mood, behavior, memory, learning, and anxiety regulation [46,47,48,49,50,51].

Taurine has been thought to be essential for the development and survival of neural cells and to protect them under cell-damaging conditions, indeed in the brain stem taurine regulates vital functions, including cardiovascular control and arterial blood pressure. Its neuroprotective effects involve also reducing neuronal apoptosis and inflammation [46], making it a subject of interest in research on neurodegenerative diseases and brain injuries and offering benefits during stroke recovery [52,53,54,55,56]. Premature infants are vulnerable to taurine deficiency because they lack some of the enzymes needed to synthesize cysteine and taurine. However, human breast milk contains high levels of taurine which is sufficient for newborns; formula milk is often supplemented with taurine, although evidence is mixed as to whether this strategy is actually beneficial or not [57,58,59,60,61,62]. Nevertheless, further studies are needed to fully understand taurine’s neurological effects.

As we will discuss below in a dedicated paragraph, taurine has been associated with several benefits especially on the cardiovascular system, including blood pressure regulation, anti-inflammatory effects, and improvements in endothelial function; overall, these properties contribute to its potential in reducing the risk of cardiovascular diseases [63,64,65].

## 3. Taurine and Cardiovascular Health

Taurine plays a crucial role in cardiovascular physiology. Numerous studies have investigated the potential cardioprotective effects of taurine, focusing on its impact on blood pressure, cardiac contractility, and vascular function. It may help reduce blood pressure in individuals with hypertension and improve endothelial function, leading to enhanced vascular health. Its antioxidant properties may also reduce the risk of cardiovascular diseases such as atherosclerosis and heart failure [66,67].

As we will see in detail in the paragraphs below, the main cardiovascular effects of taurine are attributed to a number of underlying mechanisms. For instance, its modulation of ion channels, including Ca^2+^ and potassium (K^+^) channels, influences cardiac electrical activity and vascular tone. Its role in Ca^2+^ homeostasis also impacts myocardial contractility and relaxation. Additionally, the antioxidant properties of taurine, for which the exact underlying mechanisms remain unclear, might help protect against oxidative stress, a factor involved in the pathophysiology of cardiovascular disease. Interestingly, two taurine-containing modified uridines, 5-taurinomethyluridine (τm^5^u) and 5-taurinomethyl-2-thiouridine (τm^5^s^2^u) have been identified in mitochondrial tRNA: these conjugates could be associated with the actions of taurine as an antioxidant [68,69,70,71]. Another proposed mechanism is the stabilization of intracellular levels of antioxidant enzymes like superoxide dismutase (SOD) and glutathione [72,73].

Taurine has been also implicated in metabolic regulation, particularly in relation to glucose and lipid metabolism [74,75]. Various studies indicate that taurine might help improve insulin sensitivity, making it beneficial for individuals with type 2 diabetes (T2D) or those at risk of developing the condition [76,77,78,79]. A recent preclinical study has shown that taurine can rescue pancreatic β-cell stress by stimulating α-cell trans-differentiation [80]. Additionally, taurine may aid in reducing triglyceride levels and improving lipid profiles [81,82,83,84,85], potentially lowering the risk of cardiovascular diseases and metabolic syndrome.

Preclinical investigations have provided valuable insights into the cardiovascular effects of taurine. In models of hypertension, heart failure, and atherosclerosis, taurine supplementation has consistently been shown to improve cardiac function, reduce blood pressure, and enhance vascular health. At the same time, human studies investigating taurine’s cardiovascular effects have also yielded promising results. Clinical trials have demonstrated its potential to reduce blood pressure, improve left ventricular function, and enhance exercise capacity in individuals with heart failure.

### 3.1. Taurine and Cardiac Function

Taurine accounts for ~50% of the total free amino acids in the heart; it has been shown to enhance cardiac contractility and improve heart function in both human and animal models. Animal studies have revealed that taurine deficiency induces atrophic cardiac remodeling [86], whilst taurine supplementation can increase myocardial contractility, stroke volume, and cardiac output [87,88,89,90,91,92,93]. In humans, taurine has been associated with improvements in the left ventricular function and exercise tolerance [94,95,96,97,98]. Notably, in 1985 taurine was approved as treatment for patients with heart failure in Japan [96].

The beneficial effects of taurine on Ca^2+^ and sodium (Na^+^) handling [89,90,99,100,101,102,103], myocardial energetics [104,105], and cellular signaling pathways (including glucose transport, 3-phosphoinositide-dependent protein kinase-1, AKT, sirtuin 1 (SIRT1), FOXO3, p38, NFkappaB, and others) [106,107,108,109,110,111,112,113,114,115,116,117,118,119,120,121] are thought to underlie its major cardioprotective effects. Other mechanisms include the promotion of natriuresis and diuresis, most likely via an osmoregulatory activity in the kidney, a regulation of vasopressin release, and a modulation of the atrial natriuretic factor secretion [122,123,124,125]. In addition, taurine has been shown to attenuate the actions of angiotensin II on its downstream signaling pathways, on Ca^2+^ transport, and on protein synthesis [113].

### 3.2. Taurine and Vascular Function

The endothelium, a single layer of cells lining the blood vessels, plays a crucial role in vascular health. Taurine has been shown to improve the endothelial function by promoting nitric oxide (NO) production and reducing endothelial dysfunction [126]. Enhanced endothelial function contributes to better vascular relaxation, reduced inflammation, and improved blood flow, which may benefit cardiovascular health and reduce the risk of atherosclerosis and cardiovascular events [127,128,129,130].

The ability of taurine to regulate ion channels [131,132], modulate Ca^2+^ homeostasis [133,134,135], and enhance endothelial function [136,137,138,139,140] may contribute to its antihypertensive properties. Additionally, its antioxidant activity [54,126,141,142,143] may help protect blood vessels from oxidative stress, further contributing to its beneficial effects on blood pressure regulation.

Both human and animal studies have demonstrated that taurine supplementation can lead to a modest reduction in blood pressure [144,145,146,147]. Despite the fact that the effects of taurine on a healthy endothelium remain controversial, with some investigators showing an enhancement of the endothelium-dependent relaxation in response to acetylcholine [148] and other reports not confirming these findings [145,149], its beneficial action on a dysfunctional endothelium is more consistent [130,140,144]. A synergistic action in terms of cell survival has been experimentally shown [150] when combining taurine with another well-established enhancer of vascular function, i.e., L-arginine [129,151,152,153].

Strikingly, in a recent clinical trial, 120 patients with T2D were randomly allocated to take either 1 g of taurine or placebo three times per day for an 8-week period; taurine-supplemented patients displayed a significant decrease in serum insulin and HOMA-IR (Homeostatic Model Assessment for Insulin Resistance) compared to the placebo group accompanied by a significant decline in several markers of inflammation, oxidative stress, and endothelial dysfunction [154]. A meta-analysis published in 2018 concluded that the ingestion of taurine can reduce blood pressure to a clinically relevant magnitude, without any major adverse side effects [155]. However, future studies are warranted to establish the exact effects of oral taurine supplementation on targeted pathologies and the optimal supplementation doses and periods.

### 3.3. Taurine and Athletic Performance

The presence of taurine in many energy drinks and sports supplements (~750–1000 mg in a can of 240 mL) is most likely due to its purported role in enhancing athletic performance. However, these energy drinks also contain caffeine, which has been previously linked to perceived energy boosts [156,157].

Some studies suggest that taurine may improve exercise capacity, reduce muscle damage, and alleviate exercise-induced oxidative stress. Its potential to increase muscle contractility and decrease fatigue has garnered interest among athletes. Nevertheless, conflicting findings warrant caution in interpreting these claims and several concerns on the use and abuse of energy drinks have been raised [158,159,160,161,162,163].

## 4. Taurine and Aging

### 4.1. Taurine and Longevity

Levels of taurine have been shown to decline as we age, and offsetting this loss with a taurine supplement might delay the development of age-related health problems [164,165,166,167]. Indeed, as shown in a *Science* paper recently published, when mice received taurine supplements, their lifespans increased by approximately 10% compared to the control group [168]. Mice in the taurine group also seemed healthier, with improvements in muscle endurance and strength. Researchers fed mice between 15 and 30 mg of taurine per day depending on their age. These doses would be equivalent to 3 to 6 g of taurine for an 80-kg body weight, which is within the safe limits according to European Food Safety Authority recommendations [169,170].

Taurine was also shown to shape the gut microbiota of mice and positively affect the restoration of intestinal homeostasis [171], suggesting that it could be harnessed to re-establish a normal microenvironment and to treat or prevent gut dysbiosis.

Beneficial effects on some hallmarks of aging were observed in *Caenorhabditis elegans* worms and middle-aged rhesus monkeys (*Macaca mulatta*) [172]. The taurine-fed worms lived longer and were healthier than the controls. The monkeys had lower body weights, reduced signs of liver damage, and denser bones [168].

Consistent with these data, a previous study conducted using data from the *Korea National Health and Nutrition Examination Survey* (KNHANES) had shown that taurine supplementation can decrease the cardiometabolic risk in male elderly subjects aged 75 and older [173]. Similarly, a double-blind study conducted in 24 women randomly assigned to receive taurine (1.5 g) or placebo (1.5 g of starch) for 16 weeks revealed that taurine supplementation prevented the decrease in SOD plasma levels [141], suggesting taurine as a potential strategy to control oxidative stress during the aging process.

### 4.2. Taurine and Cell Senescence

Cell senescence represents one of the fundamental mechanisms of aging [174,175]. Senescent cells are characterized by the cell cycle arrest, decreased susceptibility to apoptosis, and release of a particular set of cytokines, known as senescence-associated secretory phenotype (SASP) [176,177,178]. Despite preventing malignant transformation, accumulation of senescent cells negatively affects tissue functionality [179,180].

Multiple evidence demonstrates that the age-dependent decrease in the taurine content is associated with cell senescence. For instance, metabolomic analyses of human umbilical vein endothelial cells (HUVECs) at different passages have revealed a correlation between lower levels of taurine and HUVECs senescence [181].

In vitro, taurine mitigated replicative aging of bone marrow-derived multipotent stromal cells and restored their osteogenic differentiation potential at late passages [182]. Deletion of *Slc6a6* (sodium- and chloride-dependent taurine transporter) resulted in a drastic shortening of the lifespan of mice [168,183]; specifically, *Slc6a6* knockout mice exhibited a high expression of senescence markers p16 and p21, mirrored by a high expression of senescence-associated beta-galactosidase (SA-β-Gal) activity in the bones and liver. Treatment of *Slc6a6* knockout mice with senolytics increased their lifespan, suggesting a causative link between cell senescence and taurine deficiency [168]. In line with these results, taurine supplementation for 10 months in aged *wild type* mice led to a reduction of senescent cells by a factor of two in the brain, gut and muscle, and almost by a factor of three in the liver and fat [168]. Some investigators indicate that taurine deficiency may induce cell senescence via activation of SMAD3 and β-catenin [184].

### 4.3. Taurine and Unfolded Protein Response

Loss of proteostasis is one of the hallmarks of aging. The burden of misfolded proteins increases with age due to the accumulation of somatic mutations, dysregulation of splicing, loss of chaperone activity, and malfunctioning autophagy [174,185]. Accumulation of misfolded proteins in the endoplasmic reticulum (ER) triggers an unfolded protein response (UPR) and ER stress, eventually resulting in cell death [186].

Knockout of *Slc6a6* triggers UPR in the murine skeletal muscle, as demonstrated by unbiased RNA sequencing and by the direct measurement of ER stress-associated proteins content [183]. In drosophila, taurine’s beneficial effects on lifespan were totally abrogated by the silencing of *Erol1* or *Xbp1* genes; the products of these genes play crucial role in resolving ER stress [187]. Taurine cotreatment also prevented detrimental consequences of UPR during glucose deprivation or cisplatin toxicity [188,189].

### 4.4. Taurine and Telomere Attrition

Telomere attrition limits cell ability to proliferate endlessly [190,191,192]. The enzyme telomerase reverse transcriptase (TERT) prevents critical shortening of telomere length [174]. In vitro studies have shown that taurine can increase the TERT expression in dental-pulp-derived stem cells, thus maintaining their chondrogenic differentiation potential [193]. In line with this observation, a correlation was reported between the liver telomere length and the plasma levels of taurine in mice [194]. Taurine was also shown to mitigate detrimental consequences of telomere attrition; for instance, taurine supplementation prevented premature death of *D. rerio* with *Tert* deficiency [168].

### 4.5. Taurine and Sirtuins

Sirtuins are a family of proteins that possess either mono-ADP-ribosyltransferase or deacetylase activity [195,196]. Sirtuins regulate many signaling pathways, mostly connecting them with a metabolic state of the organism [197,198]. Their expression is decreased with age and their activation or overexpression is associated with increased longevity [199,200].

Taurine was shown to activate cytoplasmic SIRT1 in the liver, heart, and brain [121,201,202,203,204]. In these tissues, taurine-mediated upregulation of SIRT1 activity was associated with the prevention of organ dysfunction. For instance, in the heart, taurine promoted p53 inhibition via its deacetylation by SIRT1, resulting in a diminished apoptosis rate; of note, the protective effects of taurine were lost after cotreatment with a specific SIRT1 inhibitor [202].

Molecular docking modeling suggests that taurine activates SIRT1 via direct interaction with the protein; interestingly, taurine was predicted to bind another region of SIRT1 compared to the SIRT1 potent agonist resveratrol. Although the latter binds to the 289–304 amino acid sequence, taurine requires a pocket formed by amino acid 441–445 [121].

### 4.6. Taurine and Stem Cells

Depletion of stem cell pools is notably associated with aging and age-related disorders, leading to a gradual decline in organ functions and their healing capacities after damage [174,205,206,207]. Mounting data show that taurine increases the survival of stem cells, increases their regenerative capacity, and maintains stemness [208]. Notably, knocking out *Slc6a6* abrogates the development of embryonic stem cells, again pointing to the crucial role of taurine [209]. Several studies demonstrate the beneficial effects of taurine treatment on neural stem cells and stem cells involved in bone and cartilage development [193,210,211,212,213,214]; moreover, it has also been suggested that taurine may promote development of skeletal muscles [215].

## 5. Recommended Intake and Safety Concerns

Currently, there are no established dietary reference intakes (DRIs) for taurine [216]. However, it is generally believed that the typical Western diet provides sufficient taurine for most people [217,218]. Specific populations, such as vegetarians or vegans, may have a lower taurine intake, but evidence of deficiency remains limited [219,220].

The normal dietary levels of taurine can vary depending on an individual’s diet and specific food choices. Taurine is a naturally occurring amino acid found in various foods [219,221,222,223,224], including seaweed, fish, meat, and some dairy products (Table 2); the average daily intake of taurine from the typical diet is estimated to be around 40 to 400 milligrams (mg) per day in adults.

Foods that contain the highest levels of taurine come from the sea and include seaweed and shellfish; for instance, taurine represents ~80% of the total amino acid content of pacific oyster (*Crassostrea gigas*) [225].

Regarding standard supplemental doses, taurine supplements are available in various forms, including capsules, tablets, and energy drinks. The recommended dosage of taurine as a dietary supplement might vary based on the specific product and its intended use. In general, most taurine supplements are available in doses ranging from 500 mg to 2000 mg per serving. It is important to note that individual responses to dietary supplements can differ, and the appropriate dose for a person may depend on various factors, including age, weight, overall health status, and underlying medical conditions. For this reason, it is advisable to follow the recommended dosage provided on the supplement’s packaging or as advised by a healthcare professional.

Overall, taurine is considered generally safe for most individuals when consumed in moderate amounts, as found in the average diet. However, as with any dietary supplement, moderation is key, and excessive consumption of taurine supplements beyond recommended doses may lead to potential side effects, including gastrointestinal disturbances (such as nausea, vomiting, and diarrhea) and neurological symptoms (dizziness, tremors, and headache) [226,227,228]. Moreover, caution should be used because of the potential interactions between taurine supplements and certain medications, particularly those having analogous effects (e.g., lowering blood pressure), targeting similar signaling pathways (e.g., Ca^2+^, angiotensin), and used to modulate heart or central nervous system functions. medications or [49,50,229]. Pregnant and lactating women, as well as individuals with specific health conditions, such as bipolar disorder, epilepsy, or kidney problems, should exercise caution and consult healthcare professionals before taking taurine supplements.

A risk assessment study conducted by Shao and colleagues, based on toxicological evidence from several clinical trials testing taurine supplementation, established the upper level of taurine supplementation at 3 g per day [230]. The only adverse effects noted in this study after consuming a 3 g dose of taurine were gastrointestinal disorders. Notably, the minimum dose used in these trials was 3 g/day, much greater than the usual intake of taurine from a normal diet (<0.4 g/day).

## 6. Conclusions

Taurine has a diverse array of functions in human health. From its origins in animal tissues to its roles in aging, cardiovascular health, neuroprotection, and cellular function, taurine continues to capture the attention of researchers and health professionals alike. Recent findings specifically suggest that taurine is a promising cardioprotective agent, offering potential benefits for cardiovascular health in both human and animal studies. However, its role in reducing cardiovascular risk warrants further investigation, including large-scale clinical trials, making it an intriguing subject for ongoing research and potential therapeutic applications. Further research is also needed to fully elucidate its mechanisms of action and confirm its efficacy in different settings including longevity. An adequate dietary intake of taurine through a balanced diet is recommended, and caution should be exercised when considering taurine supplementation, especially at high doses.

## Figures and Tables

**Figure 1 nutrients-15-04236-f001:**
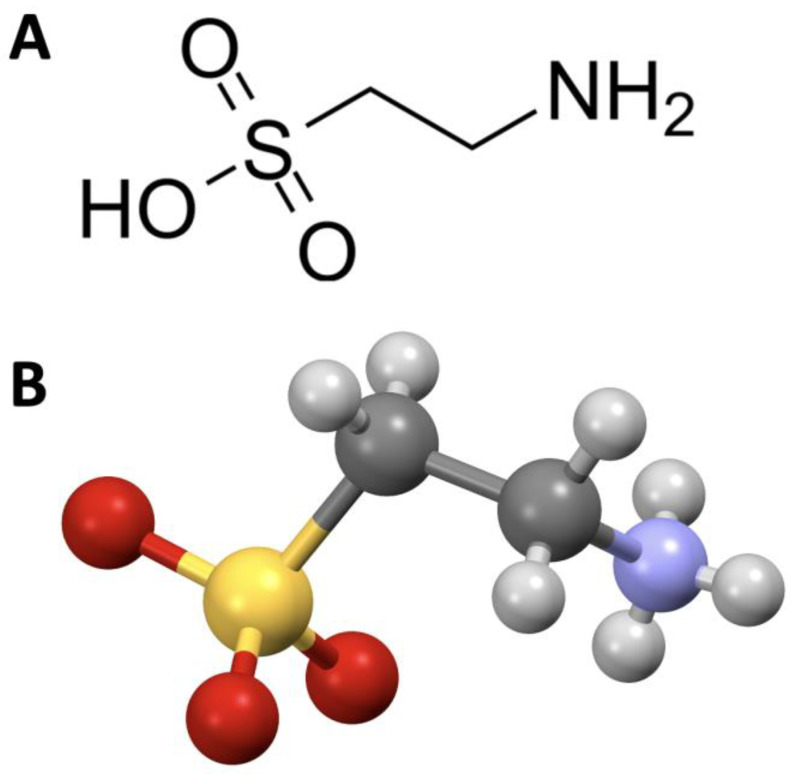
Chemical structure (**A**) and call-and-stick model (**B**) of taurine.

**Figure 2 nutrients-15-04236-f002:**
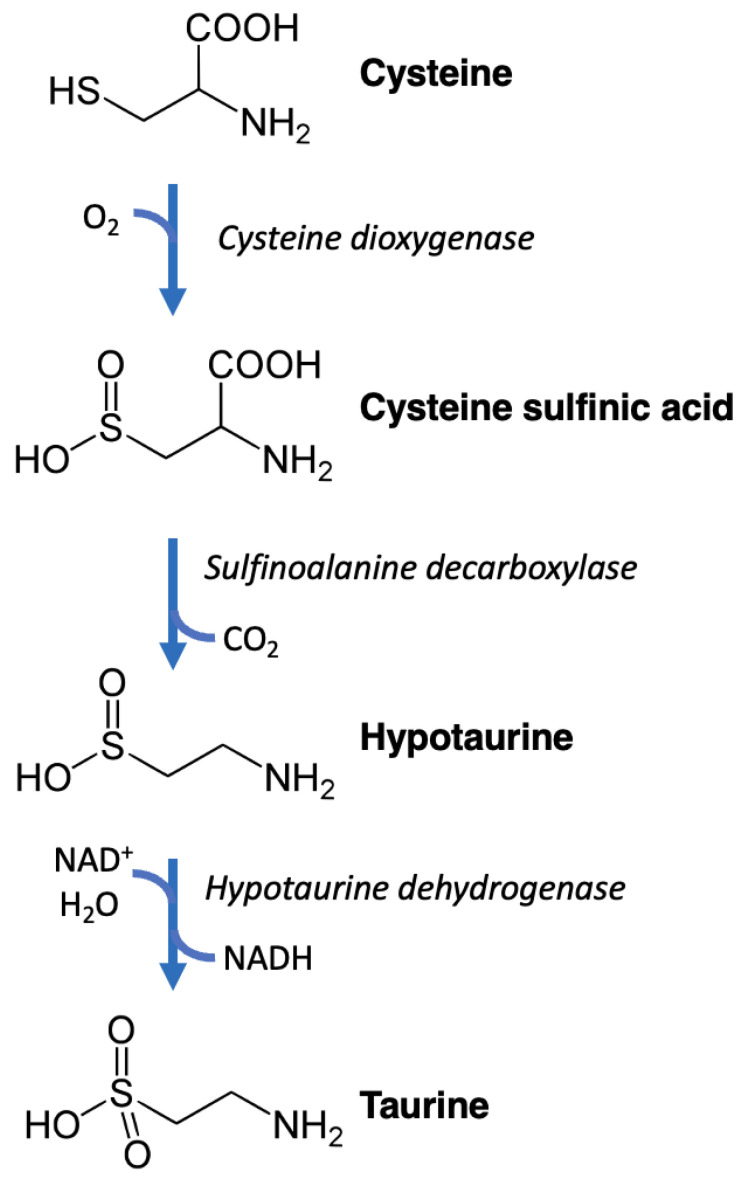
Representation of the chemical reactions of the cysteine sulfinic pathway leading to taurine synthesis.

**Table 1 nutrients-15-04236-t001:** Taurine content in human tissues (data from Refs. [7,8,9,10,11]).

Tissue	Content in μmol/L (Liquid) or μmol/g (Solid)
Bile	~200
Plasma	50–100
Leukocytes and platelets	10–50
Retina	30–40
Heart	6–25
Brain	0.8–20
Skeletal muscle	2.2–5.4
Kidney	1.4–1.8
Liver	0.3–2
Erythrocytes	0.05–0.08

**Table 2 nutrients-15-04236-t002:** Taurine content in foods.

Food	Average Content in mg/100 g
Pyropia tenera (red algae, seaweed, dried)	979
Scallops (raw)	827
Mussels (raw)	655
Porphyra haitanensis (seaweed, dried)	646
Clams (raw)	520
Oysters (raw)	507
Octopus (raw)	388
Squid (raw)	356
Turkey (raw), dark meat	306
Chicken (raw), dark meat	169
White fish (raw)	151
Pork (raw)	61
Salami (cured)	59
Ham (baked)	50
Lamb (raw), dark meat	47
Beef (raw)	43
Tuna (canned)	42
Shrimps (raw)	39
Goat’s milk (pasteurized)	6.8
Egg yolk	3.7
Yogurt	3.3
Cow’s milk (pasteurized)	2.4
Ice cream	1.9

## Data Availability

Not applicable.

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
