# Peer review of "Functional Role of Taurine in Aging and Cardiovascular Health: An Updated Overview"

_nutrients, 2023, doi:10.3390/nu15194236_

Round 1

Reviewer 1 Report

The title of this article is “Taurine and cardiovascular health: An updated overview”. This is an interesting topic, and it is an area that needs our attention. However, there are still some areas of the article that need to be revised.

1.      The focus of the article is on the cardiovascular connection of taurine. However, the author begins the article into a brief overview of taurine. In this regard, the authors need to add to the content, such as summarizing recent research advances or research trends in the link between taurine and cardiovascular health.

2.      The authors have analyzed the chemical structure of taurine, which is important. However, the authors did not go far enough in their discussion of this section, and the authors could have started with the molecular formula of taurine to explain in more detail the reasons why taurine is able to perform its beneficial effects.

3.      For the introduction of taurine and vascular function, the authors need to introduce more references or relevant experimental cases to illustrate more specifically the positive significance of taurine for the maintenance of normal vascular function.

4.      According to the general reading habits, the authors can adjust the order of the contents in Chapter 3, for example, the mechanism of action of taurine is explained before introducing the role of taurine on heart function, which is more conducive to the readers' understanding of the article.

5.      The authors describe the link between taurine and aging. For this section, the authors should have analyzed aging and cardiovascular health in conjunction with each other, pointing out more clearly the inhibitory effect of taurine on the aging of the organism and analyzing how this function is reflected in the protection of cardiovascular health.

6.      The article offers a new way of thinking about protecting cardiovascular health. However, the article lacks sufficient revelation, and the authors need to give more of their own views as well as a vision for the future.

7.      Authors are requested to carefully check the format of the references used in the article to ensure that the references are in the required format.

 Please revise the English expressions in the manuscript by removing unnecessary "the" from the sentences, making sure the sentences look more concise, and replacing words that appear too often in the text.

Author Response

The title of this article is “Taurine and cardiovascular health: An updated overview”. This is an interesting topic, and it is an area that needs our attention. However, there are still some areas of the article that need to be revised. 1. The focus of the article is on the cardiovascular connection of taurine. However, the author begins the article into a brief overview of taurine. In this regard, the authors need to add to the content, such as summarizing recent research advances or research trends in the link between taurine and cardiovascular health. R: We have added a brief introductory paragraph linking taurine and cardiovascular health 2. The authors have analyzed the chemical structure of taurine, which is important. However, the authors did not go far enough in their discussion of this section, and the authors could have started with the molecular formula of taurine to explain in more detail the reasons why taurine is able to perform its beneficial effects. R: We have elaborated the description of the structure, as requested. 3. For the introduction of taurine and vascular function, the authors need to introduce more references or relevant experimental cases to illustrate more specifically the positive significance of taurine for the maintenance of normal vascular function. R: We have expanded this section, as requested. 4. According to the general reading habits, the authors can adjust the order of the contents in Chapter 3, for example, the mechanism of action of taurine is explained before introducing the role of taurine on heart function, which is more conducive to the readers' understanding of the article. R: We have moved the description of the potential mechanisms, as requested. 5. The authors describe the link between taurine and aging. For this section, the authors should have analyzed aging and cardiovascular health in conjunction with each other, pointing out more clearly the inhibitory effect of taurine on the aging of the organism and analyzing how this function is reflected in the protection of cardiovascular health. R: We have expanded the section, as requested. 6. The article offers a new way of thinking about protecting cardiovascular health. However, the article lacks sufficient revelation, and the authors need to give more of their own views as well as a vision for the future. R: We have edited the manuscript in order to include more critical insights. 7. Authors are requested to carefully check the format of the references used in the article to ensure that the references are in the required format. R: In the revised manuscript, we have used the Endnote software using the format for the journal “Nutrients”. Please revise the English expressions in the manuscript by removing unnecessary "the" from the sentences, making sure the sentences look more concise, and replacing words that appear too often in the text. English has been revised by a native English speaker.

Reviewer 2 Report

The paper “Taurine and cardiovascular health: An updated overview”  provides an overview of the scientific literature and presents an analysis of the effects of taurine on various aspects of human health, focusing on cardiovascular pathophysiology, but also including aging, athletic performance, metabolic regulation, and neurological function.

The topic is of interest and the article is original. The arguments  are coherent with the aims of the Journal. The References are updated to the most recent research. The article is correct from a syntactic-grammatical and lexical point of view. The manuscript is well articulated and it is presented in a manner consistent with the hypotheses formulated.

I believe that it is ready to be shared with scientific community

Author Response

Thanks

Reviewer 3 Report

The manuscript offers a comprehensive review of taurine and its potential impacts on human health. The authors cover various important aspects, including taurine sources, its effects on cardiovascular health, aging, and safety considerations. In general, the manuscript provides valuable insights into this subject.

  1. In the introduction, the authors should clearly state the purpose and scope of their review. They should emphasize the need for an updated review, considering new evidence regarding taurine's role in health. This section should engage readers and set the stage for the subsequent sections.
  2. Regarding taurine sources, they should categorize them into natural and artificial. In the natural category, they should include seafood, eggs, and seaweed.
  3. While the review primarily focuses on cardiovascular health, the authors should justify why they limit it to this area despite taurine's potential benefits for various conditions like immune health, eye health, liver function, skin conditions, and mental health conditions.
  4. The discussion of taurine intake in special diets, particularly vegetarian and vegan diets, is crucial. The authors should address this topic effectively, including information on taurine-rich plant-based sources and whether taurine supplements are recommended for individuals following these diets.
  5. Safety concerns related to taurine-containing energy drinks should be elaborated upon. The authors should provide specific information on known adverse effects and their prevalence. Additionally, they should discuss regulatory measures addressing these concerns and present a balanced view of taurine's risks and benefits in energy drinks.
  6. Cultural and regional variations in taurine consumption should be highlighted. The authors should include examples of taurine-rich foods in different cultures and explore the reasons behind these dietary preferences. They may also examine whether areas with high prevalence of cardiovascular diseases have lower taurine consumption.
  7. Potential interactions between taurine supplements and common cardiovascular medications should be included. This information is essential for readers concerned about drug interactions. Specific examples of medications and their interactions with taurine would enhance this section.

Need to check and corect for all typos

Author Response

The manuscript offers a comprehensive review of taurine and its potential impacts on human health. The authors cover various important aspects, including taurine sources, its effects on cardiovascular health, aging, and safety considerations. In general, the manuscript provides valuable insights into this subject.

R: We thank this Reviewer for the words of appreciation.

In the introduction, the authors should clearly state the purpose and scope of their review. They should emphasize the need for an updated review, considering new evidence regarding taurine's role in health. This section should engage readers and set the stage for the subsequent sections.

R: We have modified the introduction, as requested.

Regarding taurine sources, they should categorize them into natural and artificial. In the natural category, they should include seafood, eggs, and seaweed.

R: We have provided a Table (Table 2) that includes all the natural sources of taurine, including seafood, eggs, and seaweed.

While the review primarily focuses on cardiovascular health, the authors should justify why they limit it to this area despite taurine's potential benefits for various conditions like immune health, eye health, liver function, skin conditions, and mental health conditions.

R: Unfortunately we do not have the sufficient expertise to deeply discuss other conditions like immune health, eye health, liver function, skin conditions, and mental health; however, we do mention some of them in our manuscript.

The discussion of taurine intake in special diets, particularly vegetarian and vegan diets, is crucial. The authors should address this topic effectively, including information on taurine-rich plant-based sources and whether taurine supplements are recommended for individuals following these diets.

R: We have included in the review the main studies on this topic, which remains highly controversial.

Safety concerns related to taurine-containing energy drinks should be elaborated upon. The authors should provide specific information on known adverse effects and their prevalence. Additionally, they should discuss regulatory measures addressing these concerns and present a balanced view of taurine's risks and benefits in energy drinks.

R: We have expanded the section, as requested, also providing references to other studies more focused on this topic.

Cultural and regional variations in taurine consumption should be highlighted. The authors should include examples of taurine-rich foods in different cultures and explore the reasons behind these dietary preferences. They may also examine whether areas with high prevalence of cardiovascular diseases have lower taurine consumption.

R: We thank this Reviewer for this suggestion. However, we did not find reliable data on these topics. The only published study that we found addressing regional variations was limited to one nation (PMID: 12908613).

Potential interactions between taurine supplements and common cardiovascular medications should be included. This information is essential for readers concerned about drug interactions. Specific examples of medications and their interactions with taurine would enhance this section.

R: We have addressed this topic. Thanks for the suggestion.

Round 2

Reviewer 3 Report

The authors have addressed my previous comments, but there is still room for improvement in the introduction. To enhance its quality, it is advisable for them to acknowledge the previous review before them and specify its completion date. Additionally, a brief summary of the updates made since that review should be provided, along with a clear justification for these updates in the context of current knowledge. Furthermore, it is essential to explicitly justify why this review focuses exclusively on cardiovascular health and aging, excluding other aspects of health, to provide readers with a well-defined scope and purpose for the manuscript.

Author Response

The authors have addressed my previous comments, but there is still room for improvement in the introduction. To enhance its quality, it is advisable for them to acknowledge the previous review before them and specify its completion date. Additionally, a brief summary of the updates made since that review should be provided, along with a clear justification for these updates in the context of current knowledge. Furthermore, it is essential to explicitly justify why this review focuses exclusively on cardiovascular health and aging, excluding other aspects of health, to provide readers with a well-defined scope and purpose for the manuscript.

R: We thank this Reviewer for her/his comment. We have already explained in the previous round that we have focused this review on aging and cardiovascular health because we do not have the necessary expertise to discuss all the aspects of health. It is unclear to which review this Reviewer is referring to: making a search in PubMed for reviews containing the terms "Taurine" "Aging" "Cardiovascular health" does not give any result (please see screenshot).
